

# How much should we believe correlations between Arctic cyclones and sea ice extent?

Jamie G. L. Rae[1], Alexander D. Todd[1,2], Edward W. Blockley[1], and Jeff K. Ridley[1]

[1]Met Office, FitzRoy Road, Exeter, EX1 3PB, United Kingdom
[2]College of Engineering, Mathematics and Physical Sciences, University of Exeter, Exeter, EX4 4QF, United Kingdom

*Correspondence to:* J.G.L. Rae (jamie.rae@metoffice.gov.uk)

**Abstract.** This paper presents an analysis of Arctic summer cyclones in a climate model and in a reanalysis dataset. A cyclone identification and tracking algorithm is run for output from model simulations at two resolutions, and for the reanalysis, using two different tracking variables (mean sea-level pressure and 850 hPa vorticity) for identification of the cyclones. Correlations between characteristics of the cyclones and September Arctic sea ice extent are investigated, and the influence of the tracking variable, the spatial resolution of the model, and spatial and temporal sampling, on the correlations is explored. We conclude that the correlations obtained depend on all of these factors, and that care should be taken when interpreting the results of such analyses, especially when the focus is on one reanalysis, or output from one model, analysed with a single tracking variable for a short time period.

## 1 Introduction

Sea ice is an important part of the climate system due to the key role it plays in the energy balance of the polar regions. In summer its high albedo reduces ocean warming, while in winter its low thermal conductivity acts to insulate the cold atmosphere from the warmer ocean below. In addition, ice melting and growth impacts the ocean temperature through heat exchange, and ocean stratification is affected through salinity changes. Arctic sea ice has undergone substantial changes since satellite-based passive microwave observations first became available nearly four decades ago. Between 1979 and 2012, the annual mean ice extent decreased on average by 3.5 to 4.1% per decade, while ice extent at the minimum of the annual cycle in September decreased by 9.4 to 13.6% per decade over the same period (Vaughan et al., 2013). The Arctic sea ice extent reached record lows in 2007 and 2012. In both years, preconditioning through thinning over several decades had made the ice more susceptible to dramatic reductions (Zhang et al., 2008; Parkinson and Comiso, 2013; Babb et al., 2016).

As well as the long-term negative trend in September Arctic sea ice extent, there is also considerable interannual variability, due to the complex interactions between a variety of physical processes acting on the ice. The September minimum Arctic sea ice extent in any given year will be influenced by seasonal and shorter-term effects, including dynamical and thermodynamic




processes in both the atmosphere and the ocean, as well as longer-term trends. Various effects are thought to have contributed to the summer 2007 record minimum, including: preconditioning (Zhang et al., 2008); large-scale atmospheric transport of heat into the Arctic (Graversen et al., 2011); anomalous oceanic heat flux through the Bering Strait (Woodgate et al., 2010); changes in cloud cover leading to increased surface and basal melting (Kay et al., 2008); and anomalous atmospheric circulation patterns

leading to increased ice motion, transpolar drift and ice flux out of the Arctic through the Fram Strait (Zhang et al., 2008).

A low ice extent also occurred in 2012, when the National Snow and Ice Data Center reported that a new record low sea ice extent was reached on 26th August. Prior to this, a strong cyclone had entered the Pacific sector of the Arctic in early August (Simmonds and Rudeva, 2012), where it had an immediate impact on the sea ice. An area of ice in the region of the Chuchki Sea and Bering Strait, measuring about $0.4 \times 10^6$ km$^2$, broke away from the main ice pack. This exposed more of the

ocean surface, leading to increased absorption of solar radiation and consequently more ice melt, and also made more of the ice vulnerable to breakup by waves, including those resulting from the storm (Parkinson and Comiso, 2013). However, without preconditioning making the ice more vulnerable to the effects of storms it is unlikely that the 2012 storm would have had the impact it did. Furthermore, the storm was not necessarily crucial to the reaching of a new record minimum: the model study of (Zhang et al., 2013) suggested that in the absence of the storm the ice extent would still have reached a new minimum in that

15   year. Storms are therefore not thought to have played a crucial role in the record Arctic sea ice minima of 2007 and 2012.

Nevertheless, cyclones are thought to have a direct impact on the ice (Kriegsmann and Brümmer, 2014). Crawford and Serreze (2016) analysed cyclones in the Modern-Era Retrospective Analysis for Research and Applications (MERRA; Rienecker et al., 2011), and found that the number of cyclones over the central Arctic peaked in summer, with many originating over Siberia. Cyclones will affect cloud cover, which will in turn have an impact on sea ice through changes to radiation and precipitation

(e.g., Eastman and Warren, 2010). Meanwhile, the surface winds associated with the cyclone are likely to affect sea ice dynamics (e.g., Ogi et al., 2010), which could cause ice to break up or be advected, leading to exposure of open water, and resulting in ocean warming and further melting in summer, or freeze-up and additional ice formation in winter. In addition, several recent studies have found apparent connections between cyclones in the Arctic during the summer and sea ice extent in September. Simmonds and Keay (2009) used the University of Melbourne cyclone identification and tracking algorithm (Simmonds et al.,

2003) with mean sea-level pressure (MSLP) fields from the JRA-25 atmospheric reanalysis (Onogi et al., 2007), and looked for correlations between the characteristics (number, depth and radius) of cyclones entering the Arctic in September (i.e. at the end of the Arctic sea ice melt season) and the September sea ice extent from the passive microwave data from the National Snow and Ice Data Center over the period 1979-2008. They considered only cyclones passing over ocean or ice points, rather than land. While they found no significant correlations of September ice extent with cyclone number, they did find significant

strong negative correlations with cyclone depth and radius, suggesting that deeper, larger cyclones later in the melt season lead to more removal of sea ice.

Screen et al. (2011) used the same algorithm and the MSLP fields from the same atmospheric reanalysis as Simmonds and Keay (2009), with sea ice concentrations from the HadISST dataset (Rayner et al., 2003). For the period 1979-2009, they found that in years in which the ice extent was at least one standard deviation less than that of the previous year (which they termed "ice

loss years"), there were fewer cyclones in the Arctic in the early part of the melt season (May-July). They suggested various



plausible mechanisms for this apparent relationship, including cloud processes, and changes in atmospheric circulation having an impact on ice drift, leading to less removal of ice during the melt season. The result was less robust when the extent was at least one standard deviation greater than the previous year's ("ice gain years"). As noted above, Simmonds and Keay (2009) and Screen et al. (2011) both used the same identification and tracking algorithm and the same reanalysis. It is likely that the

cyclone track characteristics, such as track density and mean cyclone intensity, found for a given atmospheric dataset will depend on the specific details of the algorithm used (see, e.g., Neu et al., 2013; Rudeva et al., 2014), as well as on the variable used for tracking (for example, 850 hPa vorticity or mean sea-level pressure - see Hodges et al., 2003). On the other hand, Hodges et al. (2003) applied a single algorithm to several different atmospheric reanalyses, and found that in the Northern Hemisphere the results were comparable at the synoptic scale, but different for smaller-scale features. They also suggested that

in some cases the results may depend on the spatial resolution of the reanalysis.

Here, we use a single cyclone identification and tracking algorithm with two different tracking variables to analyse Arctic cyclones for two model simulations and a reanalysis dataset, with the aim of studying correlations between cyclones and Arctic sea ice extent. In Sect. 2, we give details of the reanalysis, sea ice datasets and model simulations used, as well as the tracking algorithm. We then present our results for the cyclone characteristics and their correlations with sea ice extent in Sect. 3. In

Sect. 4, we discuss the results in the context of the sensitivity of cyclone characteristics, and their correlations with sea ice extent, to tracking variable, model resolution, and spatial and temporal sampling. We conclude in Sect. 5 by discussing the implications for studies of cyclone-ice correlations, and making some suggestions for future investigations.

## 2    Models, data and methods

### 2.1    Model output

We use output from the GC2 configuration (Williams et al., 2015) of the HadGEM3 coupled climate model (Hewitt et al., 2011). This consists of: an atmosphere component, the Met Office Unified Model (UM, Cullen and Davies, 1991; Davies et al., 2005); a land-surface component, based on the Joint UK Land Environment Simulator (JULES, Best et al., 2011); an ocean component based on a version of the Nucleus for European Modelling of the Ocean (NEMO, Madec, 2008); and a sea ice component based on a version of the Los Alamos CICE model (Hunke and Lipscomb, 2010). These communicate with each

other via the OASIS3 coupler (Valcke, 2006). The GC2 configuration incorporates Global Atmosphere configuration GA6 (Walters et al., 2017), Global Land configuration GL6 (Walters et al., 2017), Global Ocean configuration GO5 (Megann et al., 2014) and Global Sea Ice configuration GSI6 (Rae et al., 2015).

We use output from simulations at two model resolutions, which we denote by GC2-N96 and GC2-N216. GC2-N96 has an atmospheric horizontal resolution of 1.875° in longitude and 1.25° in latitude, while the atmospheric resolution of GC2-N216

is 0.833° in longitude and 0.556° in latitude. Both have 85 vertical levels in the atmosphere, and use the ORCA025 tripolar grid (which avoids a singularity at the north pole, and is nominally 0.25° resolution) in the sea-ice and ocean components, with 75 vertical levels in the ocean. Both are equilibrium simulations with greenhouse gas and aerosol forcings appropriate for the year 2000, as described by Williams et al. (2015), with the aerosol forcings varying seasonally. The CICE model configuration





is based on the zero-layer approximation of Semtner (1976), and has five ice thickness categories, as described by Hewitt et al. (2011) in their Appendix D. For each of GC2-N96 and GC2-N216, we analyse the last 100 years of a 150-year simulation to avoid transient effects during spin-up.

We perform cyclone tracking with two variables using 6-hourly fields of mean sea-level pressure (MSLP) and 850 hPa vorticity from the atmosphere component of the model, with the vorticities being calculated from the 850 hPa wind fields. For the analysis of potential correlations between cyclone characteristics and sea ice, we also use the September monthly mean sea ice extents from the sea ice component.

## 2.2 Reanalysis and observations

With the aim of assessing the cyclones in the climate model simulations against an atmospheric reanalysis, we identify and track cyclones in 6-hourly fields of MSLP and 850 hPa vorticity from the ERA-Interim reanalysis (Dee et al., 2011). While this is also model-dependent, it has been shown to compare favourably with observations (see, e.g., Screen and Simmonds, 2011; Lindsay et al., 2014). Again, the vorticity fields are calculated from the corresponding winds. We also use sea ice from the HadISST1.2 dataset (Rayner et al., 2003), which is derived from passive microwave satellite observations. For comparison with the sea ice fields calculated by the climate model, we first regrid the HadISST1.2 data from its original 1° resolution to the climate model ORCA025 grid. However, for the correlations with ERA-Interim cyclones we use September ice extents calculated directly from the HadISST ice concentration fields at 1° resolution. Because the model was run with forcings appropriate for the year 2000, we use ERA-Interim and HadISST data for the period 1990-2009 (i.e. 20 years centred on 2000) for comparison with the model. To calculate the correlations, we then use data for the 30-year period 1982-2011, which is similar (though not identical) to those used by Simmonds and Keay (2009) and Screen et al. (2011).

## 2.3 Cyclone identification and tracking algorithm

We use the TRACK objective cyclone identification and tracking algorithm (Hodges, 1999). The climate model output and reanalysis data are first preprocessed: they are converted to spherical harmonics, a "background field" (all wavenumbers below T5) is removed, and they are truncated via the removal of all wavenumbers above a certain threshold. The spherical harmonic fields are then all interpolated onto the same $2.5° \times 2.5°$ grid; these interpolated fields are used for input into TRACK. The algorithm then identifies and tracks either positive maxima or negative minima in the interpolated, truncated fields. At each 6-hour time point, the algorithm identifies all the maxima or minima above a certain threshold in the field. In the present study, we use thresholds of $10^{-5}$ s$^{-1}$ for vorticity and 1 hPa for MSLP. These thresholds are appropriate where smaller spatial scales have been removed by spectral filtering (as in this case), and allow the full life-cycle of a cylone to be captured; they have been used in previous studies (e.g., Hoskins and Hodges, 2002; Bengtsson et al., 2006). The cyclones thus identified at different time points are then linked together to form tracks. This study focuses on cyclones (as opposed to anticyclones), which correspond to positive maxima in the vorticity anomaly fields, or negative minima in the MSLP anomaly fields.

TRACK outputs details of all cyclones with a lifetime of at least two days in one hemisphere (in this case the northern hemisphere). This work focuses on cyclones passing over non-land points in the Arctic (where the Arctic is defined here as all





points north of 65°N). Thus, cyclone tracks satisfying this condition were extracted, and all others discarded. For cyclones that originate outside the Arctic, then pass into the Arctic, or vice versa, or for cyclones which pass over both land and non-land points, only the points on the track over Arctic non-land points were considered. In addition, tracks with a lifetime shorter than two days over Arctic non-land points were discarded.

## 3 Results

### 3.1 Cyclone characteristics

We assess cyclones by comparing cyclone characteristics (track count, track density and mean intensity) obtained from TRACK for modelled MSLP and 850 hPa vorticities in GC2-N96, GC2-N216 and ERA-Interim. The track count in a particular month is the total number of cyclone tracks in the domain of interest (all non-land points north of 65°N) in that month. The intensity of a cyclone at a given point on its track is taken to be the 850 hPa vorticity or central MSLP (with the background field removed as described in Sect. 2.3). The mean cyclone intensity in a gridbox for a given month is the mean intensity of all cyclones in that gridbox; the mean intensity for the whole domain is defined similarly. We consider the spatial distributions of multiannual-mean (over the 100 years of GC2 output and 30 years of reanalysis data) track densities and mean intensities (Figs. 1 to 4), as well as the frequency distributions, over the same period, of whole-domain track count and mean intensity. The frequency distributions for ERA-Interim often cover a narrower range of values than those for GC2, possibly because of the shorter time period of ERA-Interim (see Fig. 5 for an example).

The track densities from the vorticity-based analysis (Fig. 1) are generally higher than those from the MSLP-based analysis (Fig. 2), and the cyclones in the latter are mainly restricted to the peripheral seas in the eastern Arctic. The halo seen around Greenland in Fig. 1 occurs because the surface pressure over much of Greenland is lower than 850 hPa (due to the high orography), so there is no 850 hPa vorticity there; we therefore treat the results in this region with caution. Cyclone track densities obtained from both vorticity and MSLP are significantly lower in GC2-N96 than in ERA-Interim (Figs. 1 and 2; hatching denotes areas where a Welch t-test showed the difference to be significant at the 95% level). In the case of vorticity, this is the case mainly over the East Siberian and Laptev Seas (notably so in June - results for individual months not shown here); we ignore the apparently-significant differences in the Davis Strait and Baffin Bay because of the orography-related issues with the 850 hPa vorticity field over Greenland. Differences between GC2-N216 and ERA-Interim are mostly insignificant for both tracking variables - GC2-N216 generally gives a similar representation of cyclone track density to ERA-Interim (except over the East Siberian and Laptev Seas in June in the case of vorticity).

To explore similarities and differences between the cyclones in ERA-Interim, GC2-N96 and GC2-N216, we use a two-sample Kolmogorov-Smirnov test to determine whether the frequency distributions of track counts and intensities from each can be said to be different at 95%, 99% and 99.9% confidence levels. For each model/reanalysis and each month between May and September, the frequency distributions of whole-domain track count from the vorticity-based and MSLP-based analyses were found with 99.9% confidence to be different. For both tracking variables, the Kolmogorov-Smirnov test suggested that the track count distributions in GC2-N96 and GC2-N216 are different (95% confidence). The same was true for GC2-N96





and ERA-Interim (99% confidence for MSLP; 95% confidence for vorticity, except in August where the possibility that the distributions may be the same could not be rejected). However, we cannot say with 95% confidence that the track count distributions from GC2-N216 and ERA-Interim are different; this is the case for both tracking variables, and is consistent with the results in Figs. 1 and 2, where the differences between ERA-Interim and GC2-N216 were seen to be mainly insignificant.

While the vorticity associated with an individual cyclone is related to the MSLP at its centre, there is no simple way to relate Arctic-wide mean MSLP-based and vorticity-based intensities. Additionally, the MSLP-based tracking method is biased towards large spatial scales, and the vorticity-based method towards smaller scales (Hoskins and Hodges, 2002); the two methods thus tend to identify different systems. It is therefore difficult to compare directly mean intensities from one method with those from the other, and we do not attempt to do so. We can, however, compare the intensities obtained from the three

models. The mean cyclone intensities from both tracking variables are significantly less in GC2-N96 than in ERA-Interim almost everywhere (Figs. 3 and 4). In the case of vorticity this is true in all months between May and September, while in the case of MSLP the differences are smaller towards the end of the melt season (not shown). For both vorticity and MSLP, the mean intensity in GC2-N216 is also less than in ERA-Interim, but the differences are smaller, and are significant over a smaller area, than in GC2-N96. The results of the Kolmogorov-Smirnov test for the frequency distributions of mean intensity

were similar to those of track count. For both tracking variables, the test suggested that the distributions of mean intensity from GC2-N96 and GC2-N216 are different (99.9% confidence), as are those from GC2-N96 and ERA-Interim (99% confidence for MSLP; 95% confidence for vorticity, except in August where we cannot reject the possibility that the distributions may be the same). For MSLP-based tracking, we cannot say with 95% confidence that the mean intensity distributions from GC2-N216 and ERA-Interim are different. For vorticity-based tracking, the distributions of mean intensity from GC2-N216 and GC2-N96

were found with 95% confidence to be different in May, July and September; however, in June and August we cannot reject the possibility that the distributions may be the same. Thus, the two climate model simulations (GC2-N96 and GC2-N216), identical except for spatial resolution, generate different cyclone characteristics, while two independent models (GC2-N216 and ERA-Interim) with different resolutions can produce similar cyclones.

## 3.2   Sea ice

Before considering the impact of cyclones on Arctic sea ice, it is important that we assess the sea ice extent in the model. In this section, we compare modelled sea ice extent against that from the HadISST1.2 observationally-based dataset (Rayner et al., 2003). GC2-N96 reproduces the observed Arctic ice extent well in most months, although GC2-N216 performs better in August and September (Fig. 6). At both atmospheric resolutions, the model underestimates September mean sea-ice concentration in the Atlantic sector of the Arctic, while there are some regions of overestimation in the Pacific sector, which are more extensive

at lower atmospheric resolution (not shown here). There is less ice off the coast of Siberia at higher atmospheric resolution than at lower resolution. These differences were found to be significant at the 95% level. A more detailed evaluation of the sea ice in GC2-N96 and GC2-N216 was presented by Rae et al. (2015).





### 3.3 Cyclone-ice correlations

To explore possible links between cyclones and sea ice, we calculated Pearson correlation coefficients, and the associated p-values, between track count in each month between May and September, and September mean Arctic sea ice extent. In Fig. 7, we give the correlation coefficients for GC2-N96 and GC2-N216, and for ERA-Interim cyclones and HadISST1.2 sea ice. Results are shown only where the confidence level was at least 90% (i.e. $p \leq 0.1$). We found a positive correlation between ERA-Interim vorticity-based track count in the early part of the melt season (May-June) and September mean HadISST1.2 ice extent (suggesting that more cyclones in May-June result in a larger ice extent in September). However, the equivalent correlation for MSLP-based track count was not significant. In addition, in GC2-N216, there was a negative correlation between June MSLP-based track count and September mean ice extent (linking more cyclones in June with a smaller ice extent in September). Meanwhile, in August, towards the end of the melt season, when cyclones may be expected to play a role in breaking up the ice, and where we may expect to see a negative correlation of track count with September ice extent, we found positive correlations in both ERA-Interim/HadISST1.2 and GC2-N216, for MSLP-based cyclones.

In the later part of the melt season, we found negative correlations for mean cyclone intensities in September (GC2-N96 vorticity-based cyclones, and ERA-Interim MSLP-based cyclones), in August (GC2-N216 MSLP-based cyclones), and in July (GC2-N96 MSLP-based cyclones). However, we also found a strong and significant positive correlation between August mean ERA-Interim vorticity-based intensity and September HadISST1.2 ice extent. Earlier in the melt season, there is a positive correlation found between ERA-Interim MSLP-based intensity in May and September HadISST1.2 ice extent, and a similar positive correlation for the ERA-Interim vorticity-based intensity in June. However, in GC2-N96 there is a correlation of the opposite sign (i.e. negative) between May vorticity-based intensity and September mean ice extent.

## 4 Discussion

### 4.1 Consideration of cyclone-ice correlations in the context of previous studies

While some of the correlations we found between cyclone characteristics and September ice extent are consistent with results published by other authors (Simmonds and Keay, 2009; Screen et al., 2011), there are others that cannot be explained in relation to those studies. In addition, in some cases where based on previous work one would expect to see correlations, no such significant correlations were found, or the correlations had the opposite sign to that expected. For example, the lack of correlation between ERA-Interim MSLP-based track count in the early part of the melt season and September mean HadISST1.2 ice extent contradicts the results of Screen et al. (2011), despite the strong, positive correlation seen for the equivalent vorticity-based track count. The negative correlations seen in some cases between cyclone intensity later in the melt season (in July, August and September) and September ice extent are consistent with the results of Simmonds and Keay (2009), who found a strong and significant correlation between mean cyclone depth in September and mean September ice extent, where they defined cyclone depth as the pressure difference between the centre and edge of the cyclone. This is contradicted by the positive correlation seen for August mean intensity in ERA-Interim. Additionally, the negative correlation between May vorticity-based



intensity and September mean ice extent in GC2-N96 tends to contradict the results of Screen et al. (2011). On the other hand, the positive correlation found between ERA-Interim MSLP-based intensity in May, and September HadISST1.2 ice extent could be consistent with the findings of Screen et al. (2011), although their focus was on track count rather than cyclone intensity. In the rest of this section, we attempt to explain these findings, and our results in general, by considering the impact

of differences in the model simulation, choice of tracking variable, and spatial and temporal sampling, on the correlations obtained.

## 4.2 Dependence on model and resolution

For a given tracking variable (vorticity or MSLP), we saw wide variations in cyclone-ice correlations between the models. The track densities and mean intensities in GC2-N96 are significantly lower than those in GC2-N216, suggesting that these

are strongly resolution-dependent, as the model setups for those simulations were identical apart from the resolution. These differences in cyclone characteristics may lead to differences in the interactions between cyclones and sea ice, and thus to the different correlations that we saw in those simulations. Meanwhile, despite the cyclone characteristics in GC2-N216 being similar to those in ERA-Interim, the correlations with September ice extent are different. This is likely to be because outputs are from different models: the ERA-Interim data are from an atmosphere-surface-wave model that assimilates observations

(Dee et al., 2011), whereas GC2-N216 is a fully-coupled climate model without data assimilation. In addition, ERA-Interim and HadISST1.2 include the effects of climate change, whereas GC2-N96 and GC2-N216 are equilibrium climate model runs. It is thus likely that other factors are having an influence. Model resolution, and other model properties, can therefore play a potentially-crucial role in determining the correlations seen.

As mentioned in Sect. 2.3, the mean cyclone intensities presented in Fig. 3 have had the background field (wavenumbers $< 5$)

removed, and can thus be thought of as anomalies. To evaluate the effect of the removal of the background field, we also plotted maps of absolute intensity (not shown here). For the vorticity-based analysis, the intensity obtained from the ERA-Interim data was intermediate between those from GC2-N96 and GC2-N216, suggesting that resolution may be more important when absolute values are considered (the $1°$ resolution of ERA-Interim is intermediate between those of the other two simulations). In the MSLP-based analysis, however, the absolute intensities from ERA-Interim do not lie between those of the two GC2

simulations, implying that the situation is more complicated than simple dependence on resolution. The differences between geographical distributions of absolute intensities and intensity anomalies, and their dependence on resolution and tracking variable, is also reflected in differences in cyclone-ice correlation depending on whether absolute values or anomalies are used. We have presented correlations only for intensity anomalies, as we believe the departure of the intensity from the background field to be a more meaningful predictor of the possible impact of the cyclone on the sea ice.

## 4.3 Dependence on tracking algorithm and variable

A number of differences were seen between the cyclone-ice correlations for vorticity-based and MSLP-based track counts. For example, we found a positive correlation between HadISST September ice extent and track count early in the melt season (May and June) for vorticity-based cyclones in ERA-Interim, but no similar correlation with the MSLP-based track count. This is



in contrast to Screen et al. (2011), who used MSLP from the JRA-25 reanalysis as their tracking variable and found apparent links between early-melt-season track count and September ice extent. However, as well as a different reanalysis dataset, Screen et al. (2011) used the University of Melbourne cyclone finding and tracking algorithm (Simmonds et al., 2003), rather than the TRACK algorithm (Hodges, 1999) applied here. Neu et al. (2013) applied several different tracking algorithms to the

same atmospheric reanalysis and examined a variety of cyclone characteristics, including track count and cyclone intensity. They found wide variations in track count between the algorithms, depending on such factors as: the threshold for detection; the minimum distance between two cyclones; and whether the input data were preprocessed by smoothing (which has the same effect as reducing the resolution, leading to fewer cyclones being detected). These variations have in some previous studies been found to be substantial enough that two different algorithms give opposite signs for the trends in cyclone characteristics

in particular regions under climate change (Raible et al., 2008). Thus, the lack of consistency with the results of Screen et al. (2011) may not be surprising, and this provides a good illustration of the potential for different algorithms to give different results.

In the present study, tracking performed on the MSLP field yielded fewer cyclone tracks than that on the vorticity field, although the dependence on resolution was similar for both variables. Some of the algorithms in the study of Neu et al. (2013)

used vorticity as the tracking variable, some used MSLP, some used a combination of the two, while some used other variables, such as 850 hPa geopotential height. Neu et al. (2013) did not draw conclusions about the impact of tracking variable on cyclone characteristics; they emphasised the difficulty of attributing differences in cyclone characteristics to specific aspects of the algorithms, due to multiple differences between the algorithms, which are likely to combine non-linearly. Rudeva et al. (2014) did investigate sensitivity of cyclone characteristics to particular aspects of the algorithms, but not to the variable used.

However, Hodges et al. (2003) used the identification and tracking algorithm of Hodges (1999) to analyse cyclone tracks in several reanalysis datasets using both vorticity and MSLP as tracking variables. As in the present study, they detected fewer cyclones with MSLP than with vorticity, which they attributed to the fact that MSLP-based analyses tend to pick up larger-scale features than vorticity-based analyses, leading to fewer detections in regions where smaller-scale features dominate. In addition, we found that MSLP-based cyclones were concentrated in the eastern Arctic to a greater extent than vorticity-based

cyclones (compare Figs. 1 and 2), which is also consistent with the results found by Hodges et al. (2003) for winter (see their Fig. 1). Given these differences in the number of cyclone tracks and their geographical distribution between the two tracking variables, it is perhaps not surprising that we also see such differences in the correlations with September ice extent. This underlines the possibility for the same algorithm to give different results depending on the variable used.

There are also differences in the intensity-ice correlations for the two tracking variables. There is a negative correlation

between MSLP-based mean intensity from ERA-Interim late in the melt season (September) and HadISST1.2 September ice extent, consistent with Simmonds and Keay (2009), but no such correlation for the vorticity-based intensity. We also found negative correlations between September ice extent and MSLP-based cyclone intensity in August for GC2-N216 and in July for GC2-N96, and vorticity-based intensity in GC2-N96 in May and September. However, for the reasons given in Section 3.1 we were unable to compare directly the mean intensities from the two methods.



## 4.4 Impact of domain choice

In the preceding analysis, we followed Simmonds and Keay (2009) in considering only cyclones passing over non-land points north of 65°N. However, other authors (e.g., Screen et al., 2011) have included all points (land, ocean and ice) in that region. We therefore examined the impact of this spatial sampling by recalculating the correlations, between September ice extent and cyclone characteristics in preceding months, using all cyclones north of 65°N (not shown here). Some correlations are significant in both domains. For example, for the ERA-Interim data, the correlations for the track count from the vorticity-based analysis in May, and the intensities from the MSLP-based analysis in May and the vorticity-based analysis in June, are strong and significant for both domains. Similarly, for GC2-N96, the correlations for vorticity-based intensities in May and September, and MSLP-based intensity in July, are significant in both domains. Finally, for GC2-N216, the correlations for track count and intensity in August are significant in both domains. However, other correlations were found to be significant in only one of the domains, suggesting that the results are at least partly domain-dependent.

## 4.5 Impact of temporal sampling

In the preceding analysis, we used 30 years of data for ERA-Interim/HadISST1.2 but 100 years for GC2-N96 and GC2-N216. We now investigate the effect of shorter temporal sampling by calculating the correlations, over non-land points north of 65°N, for different, discontinuous, 30-year periods (the first, middle and last 30 years) within the 100 years of GC2-N96 and GC2-N216 output. Fig. 8 shows these, and compares them with the correlations over the whole 100 years. None of the correlations are significant at the 90% level in all of the 30-year periods. Most of the correlations that are significant at either 90% or 95% confidence over the full 100-year period are significant in only one of the 30-year subsets; meanwhile, some significant correlations seen in one of the 30-year periods are found not to be significant over the whole 100 years. The correlation for vorticity-based mean intensity in September in GC2-N96, found to be significant at the 90% level over the whole 100 years, is not significant in any of the 30-year subsets. For GC2-N96, significant positive correlations of August track count with September ice extent were found for two of the 30-year periods (95% confidence in the second; 90% in the third), and for the whole 100-year period (95% confidence).

The potential for identification of spurious correlations is illustrated by the May intensity from the MSLP-based analysis of GC2-N216 model output. In this case, the correlation with September sea ice extent was positive in the first 30-year period (95% confidence), and negative in the third (90% confidence), while over the whole 100 years there was no correlation significant at 90% confidence or above. Thus, different, discontinuous, 30-year periods of a 100-year time series can produce a significant positive correlation, a significant negative correlation, or no significant correlation at all, highlighting the dependence of the results on temporal sampling.

Studies using reanalyses for cyclone tracking, and satellite-based observations for sea ice concentration are necessarily limited by the availability of satellite data to the period since 1979. So, for example, Simmonds and Keay (2009) therefore considered the period 1979-2008, Screen et al. (2011) considered 1979-2009, and we have considered 1982-2011 in our anal-

ysis of the ERA-Interim data in the present paper. However, the results presented here suggest that the correlations obtained in such analyses may be dependent on the period selected.

## 5   Summary and conclusions

We have used a single cyclone identification and tracking algorithm with two different tracking variables (850 hPa vorticity and
MSLP), and three model simulations (ERA-Interim reanalysis, and two simulations with the same climate model at different atmospheric resolutions) to study the number of cyclones in the Arctic during the summer sea ice melt season, and their mean intensity. We also studied the correlations between these cyclone characteristics and the September mean Arctic sea ice extent. We found some correlations between September sea ice extent and cyclone characteristics that are consistent with previous studies, and others that are not.
Crucially, the correlations were found to be dependent on various aspects of the model, such as resolution, as well as on the variable used for tracking, and on spatial and temporal sampling. One key result for the correlation between MSLP-based mean cyclone intensity in May, and sea ice extent in September, showed significant positive and negative correlations for discontinuous 30-year subsets of the same 100 years of output from a particular model simulation, while over the full 100 years the correlation was not significant. For this reason, we suggest that caution should be exercised when performing studies
such as this, especially where data are only available for a limited period. The interaction between cyclones and sea ice is clearly complicated, involving many competing physical processes, and further investigations, focused on developing a better understanding of these processes, would be beneficial.

*Code and data availability.*   For details of how to obtain code for the GC2 configuration of HadGEM3, please see Williams et al. (2015). Details of the TRACK algorithm and how to obtain it can be found at http://www.nerc-essc.ac.uk/ kih/TRACK/Track.html. Information
about the ERA-Interim reanalysis dataset and how to obtain it is given by Dee et al. (2011). The HadISST sea ice data are available at http://www.metoffice.gov.uk/hadobs/hadisst/.

*Competing interests.*   The authors declare no competing interests

*Acknowledgements.*   We are grateful to Kevin Hodges of the Department of Meteorology, University of Reading, and Ruth McDonald and Claudio Sanchez of the Met Office, for advice and assistance on running the TRACK algorithm. This work was supported by the Joint UK
BEIS/Defra Met Office Hadley Centre Climate Programme (GA01101)



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




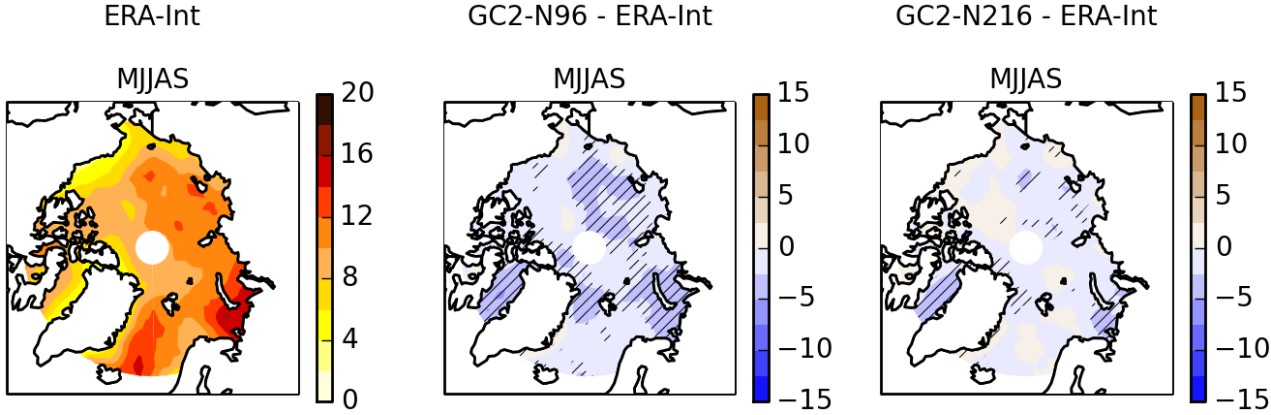

**Figure 1.** Vorticity-based tracking: 1990-2009 MJJAS mean ERA-Interim cyclone track density ($10^{-6}$ km$^{-2}$ month$^{-1}$), and differences between this and the track density from climate model runs GC2-N96 and GC2-N216 (MJJAS mean over last 100 years of 150-year run). Hatching indicates that the difference is shown by a Welch t-test to be statistically significant at the 95% level.

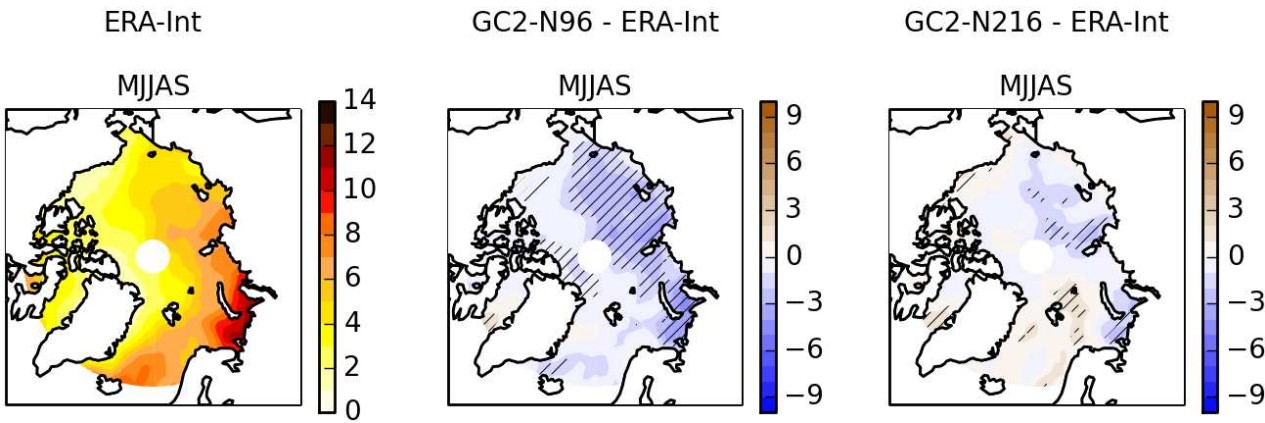

**Figure 2.** MSLP-based tracking: 1990-2009 MJJAS mean ERA-Interim cyclone track density ($10^{-6}$ km$^{-2}$ month$^{-1}$), and differences between this and the track density from climate model runs GC2-N96 and GC2-N216 (MJJAS mean over last 100 years of 150-year run). Hatching indicates that the difference is shown by a Welch t-test to be statistically significant at the 95% level.





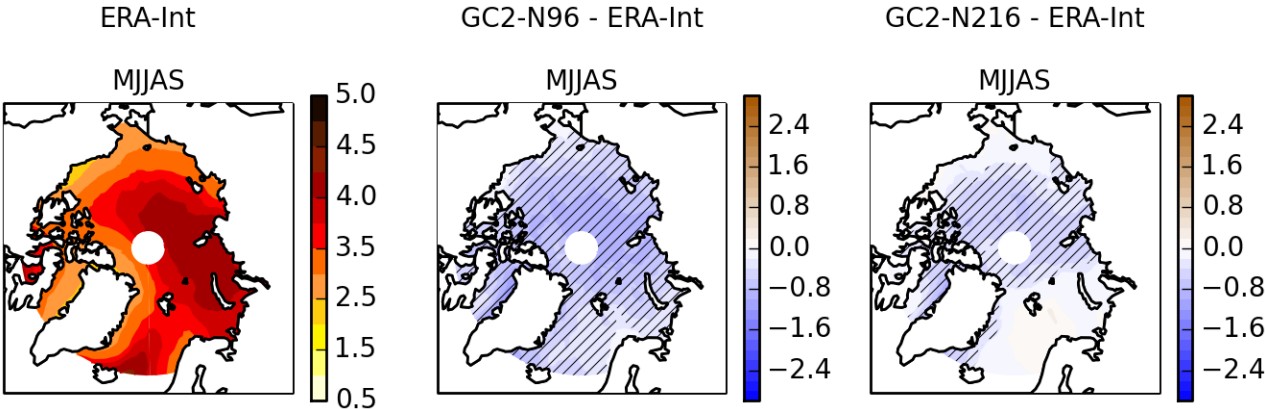

**Figure 3.** Vorticity-based tracking: 1990-2009 MJJAS mean ERA-Interim mean cyclone intensity with background field removed ($10^{-5}$ s$^{-1}$), and differences between this and the mean cyclone intensity from climate model runs GC2-N96 and GC2-N216 (MJJAS mean over last 100 years of 150-year run). Hatching indicates that the difference is shown by a Welch t-test to be statistically significant at the 95% level.

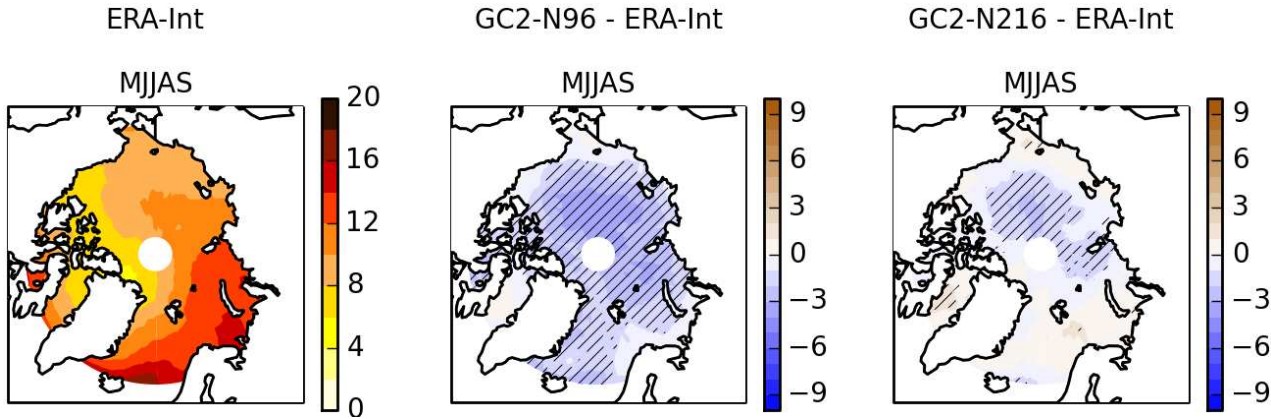

**Figure 4.** MSLP-based tracking: 1990-2009 MJJAS mean ERA-Interim mean cyclone intensity with background field removed (hPa), and differences between this and the mean cyclone intensity from climate model runs GC2-N96 and GC2-N216 (MJJAS mean over last 100 years of 150-year run). Hatching indicates that the difference is shown by a Welch t-test to be statistically significant at the 95% level.





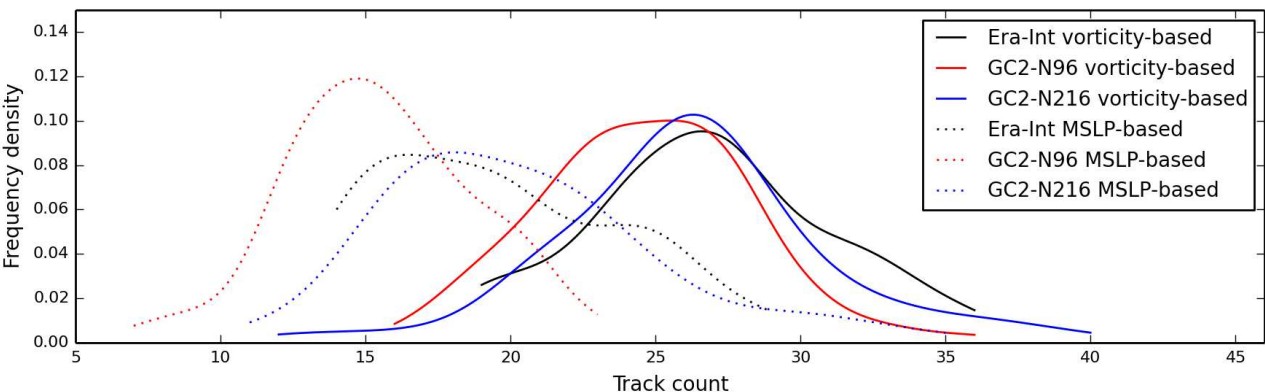

**Figure 5.** Frequency distributions of August cyclone track count. Use of the MSLP field in the tracking algorithm gives fewer cyclones than use of the vorticity field. The GC2-N216 climate model run is seen to give a similar number of cyclones to the ERA-Interim reanalysis (supported by a two-sample Kolmogorov-Smirnov test), while the lower-resolution GC2-N96 model run gives fewer cyclones.

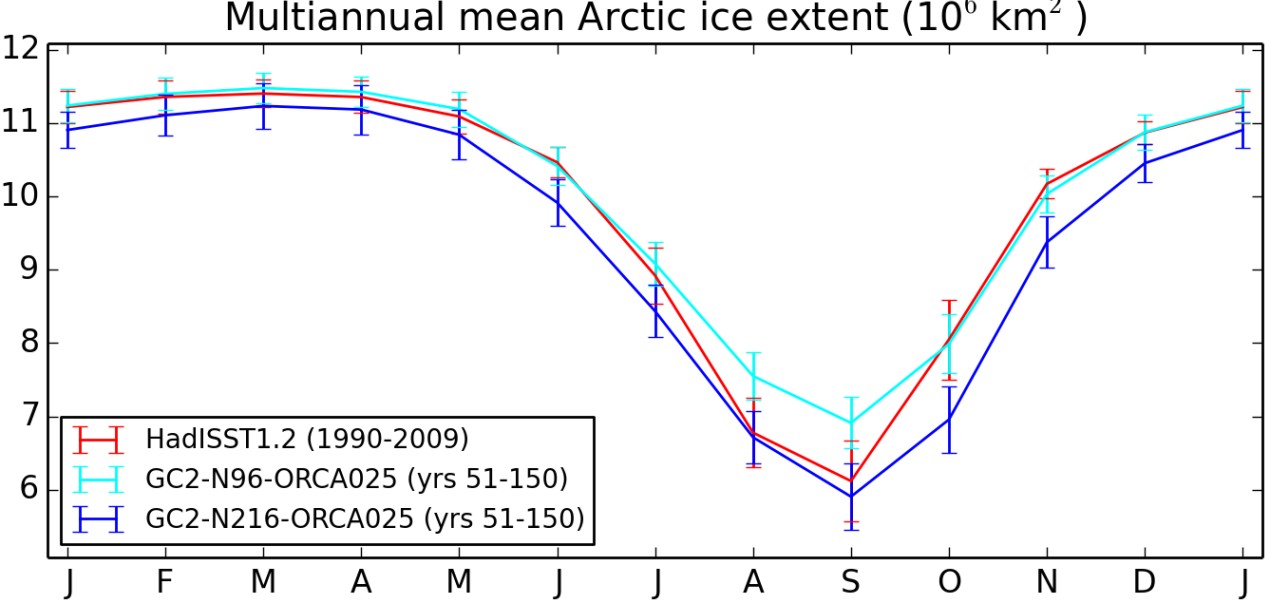

**Figure 6.** Sea ice extent multiannual-mean seasonal cycles from HadISST1.2 observational dataset, and GC2-N96 and GC2-N216 climate model runs. Error bars represent standard deviations on the multiannual means for each month, calculated over the same years as the means themselves. The ice extents in GC2-N216 and HadISST1.2 are seen generally to be within one standard deviation of each other.



| | | May | Jun | Jul | Aug | Sep |
|---|---|---|---|---|---|---|
| **Track count** – Vorticity | ERA–Int | 0.40 | 0.32 | | | |
| | GC2–N96 | | | | | |
| | GC2–N216 | | | | | |
| **Track count** – MSLP | ERA–Int | | | | 0.33 | |
| | GC2–N96 | | | | | |
| | GC2–N216 | | –0.21 | | 0.21 | |
| **Intensity** – Vorticity | ERA–Int | | 0.41 | | 0.44 | |
| | GC2–N96 | –0.24 | | | | –0.18 |
| | GC2–N216 | | | | | |
| **Intensity** – MSLP | ERA–Int | 0.39 | | | | –0.33 |
| | GC2–N96 | | | –0.21 | | |
| | GC2–N216 | | | | –0.21 | |

**Figure 7.** Pearson correlation coefficients between May-September cyclone characteristics (over ocean and ice points only) and September mean sea ice extent. Dark red squares denote positive correlations significant at 95% confidence level; light red squares denote positive correlations significant at 90% confidence level; dark blue squares denote negative correlations significant at 95% confidence level; and light blue squares denote negative correlations significant at 90% confidence level. Correlation coefficients are not shown where the confidence level is less than 90%.



**Figure 8.** Impact of temporal sampling, demonstrated by correlations over 100-year period and different, discontinous, 30-year subsets of output from GC2-N96 and GC2-N216 runs. Colours denote sign of, and confidence in, Pearson correlation coefficients between May-September cyclone characteristics (over Arctic non-land points) and September mean sea ice extent. Dark red squares denote positive correlations significant at 95% confidence level; light red squares denote positive correlations significant at 90% confidence level; dark blue squares denote negative correlations significant at 95% confidence level; and light blue squares denote negative correlations significant at 90% confidence level. White squares indicate correlation is not significant at 90% confidence or higher. Columns marked '1': first 30 years of 100-year period. Columns marked '2': middle 30 years of 100-year period. Columns marked '3': last 30 years of 100-year period.