# Peer review of "How much should we believe correlations between Arctic cyclones and sea ice extent?"

_The Cryosphere, 2017_

## Referee Comment (RC1) · Anonymous Referee #1 · 25 Sep 2017

This paper has tried to correlate the sea ice anomalies with the summer cyclonic activity in Arctic and found that the correlations are highly sensitive to the model and resolution used for tracking the cyclonic activity. While the paper is generally well written and the authors explain well their methodology, I do not recommend to accept this paper for publication in TC because this subject has already been discussed in many papers (as well presented by the authors) and that this paper only shows that the variability of sea ice extent is definitely NOT driven by the summer cyclonic activity! The correlations found (< 0.44) are not enough relevant to justify a paper in TC and obviously explains why the results are very sensitive to the model and resolution used. If the correlations would be scientifically robust, the sign of the correlations should be for example every time the same which is not the case here. Correlating sea ice extent

with summer T850 will give likely better correlations but it is out of the scope of this paper. Sorry to not be more positive...

---

## Author Comment (AC1) · 27 Sep 2017

We thank the reviewer for his/her comments. However, we respectfully suggest that he/she may have misunderstood the purpose of the paper. The aim of the paper was not simply to find correlations between sea ice extent and cyclones, but rather to analyse the robustness or otherwise of such correlations.

We showed that the correlations are dependent on spatial resolution, tracking variable, spatial domain, and time period used. We thus concluded that they are NOT robust, and that the results of previous studies that have used correlations between summer storms and Arctic sea ice may also NOT be robust.

This misunderstanding suggests that the paper in its current form may be ambiguous

with regard to its purpose and conclusions. If that is the case, we are happy to revise the wording to make it clearer.

---

## Referee Comment (RC2) · Anonymous Referee #2 · 18 Oct 2017

This paper investigates correlative relationships between Arctic cyclones and September sea ice extent using two different cyclone tracking variables from three different sources (output from two climate model runs and from one reanalysis dataset) . The results show that different tracking variables, model resolution and space/time comparisons can show contrasting cyclone/ice relationships, thereby emphasizing that caution is required when analyzing and interpreting such comparisons (e.g., as previously presented in the literature). This caution is noteworthy and helpful. Thus, this paper is deemed appropriate for TC after considering a few minor suggested revisions as listed below.

P2, L9, Suggest changing 'main ice pack' to 'main pack ice'.

P3, L3, Consider starting a new paragraph here.

[Figure]

P3, L12, Consider rephrasing the following: 'aim of studying correlations between cyclones and Arctic sea ice extent' given Reviewer #1's comment. In other words, specifically state the overall aim of this paper, which is to show how correlations (between cyclones and sea ice extent) depend on sea ice extent, tracking variable, model resolution, and time/space windows used to make the comparison (i.e., what is then stated further down on L15-16).

P5, after L4, Consider adding another short paragraph here describing the statistics used to test differences and compute correlations.

P5, L15, Since the first reference to a figure is here, consider moving Fig 5 to Fig 1 and adjust the others accordingly.

P5, L30-31, This seems like a general statement summarizing all model/reanalysis comparisons, but then it differs from the last two statements in that paragraph. Reword the first statement to clearly distinguish it from the other points being made (and/or create a table listing these results).

P6, L14, Consider starting new paragraph here.

P8, L4, fix typo: 'thesefindings'

Figs 1-4, it would be helpful and of interest to see the difference maps between GC2-N96 and GC2-N216. (This would also be helpful for interpreting the contrasting results between the 2 model runs as presented in Fig 7.)

---

## Editor Comment (EC1) · J.-L. Tison (Editor) · 19 Oct 2017

Dear Authors,

Clearly, the feedback of the two reviewers are quite contrasted. There is however a common feeling that the data seriously lack originality, which is a major criteria for acceptance in The Cryosphere.

I would therefore suggest that you provide me with:

a) clear and strong rebuttal letter showing that there is "originality" in the data and that they provide the scientific community with a clear step forward

b) a new version of the manuscript taking the reviewers comments into account, with

both marked changes in a "track change" version of it, and a "clean" version

As clearly stated in "The Cryosphere Discussion" rules, this does not however at all guarantee that it will grant access to final publication in "The Cryosphere".

For your information, I will be in the field from November 14th to January, 3.

Best regards,

Jean-Louis Tison

---

## Editor Comment (EC2) · J.-L. Tison (Editor) · 9 Nov 2017

Dear co-authors,

I see that you have replied individually to the reviewers comments.

Would you be kind enough to provide me with the documents I asked in my previous comment of Octobre 15?... It will be easier for me to take a final decision..

Apologies if you have already done this in the two last days,

Best regards,

Jean-Louis Tison

---

## Author Comment (AC5) · 9 Nov 2017

Dear Professor Tison.

Please find attached the documents you requested.

Jamie Rae

Please also note the supplement to this comment:
https://www.the-cryosphere-discuss.net/tc-2017-140/tc-2017-140-AC5-supplement.pdf
* * *

---

## Author Response (AR1)

Met Office,
FitzRoy Road,
Exeter,
EX1 3PB,
United Kingdom.

7[th] November, 2017

Dear Professor Tison,

We thank both you and the reviewers for your comments on our paper *How much should we believe correlations between Arctic cyclones and sea ice extent?*. We have already uploaded a short response to the concerns expressed by Reviewer #1, but give a more detailed rebuttal below.

As Reviewer #1 points out, a number of previous papers have studied correlations between Arctic cyclones and sea ice extent in observations and reanalyses. Indeed, we cited some of these studies in our paper. However, the fact that these studies used observations/reanalyses, rather than output from a coupled GCM, meant that they were restricted to the ~30 years of available satellite observations of sea ice extent. In addition, they used only one resolution and one tracking variable. These studies found apparent correlations/links between summer cyclones and September ice extent.

In contrast, in our paper we use long (100-year) runs of a coupled climate model, at two different resolutions, analysed with two different tracking variables. In doing so, we show that such correlations are likely to be dependent on resolution, tracking variable, spatial domain, and time period used, and conclude that they are therefore not robust. This lack of robustness, and its implications for the correlations apparently found by other authors, is the key result of our paper, and our main conclusion is that this should be taken into account when interpreting the results of those previous studies. **Our analysis of the dependence of correlation on resolution, tracking variable, spatial domain, and time period used is original. In addition, the conclusion that this dependence has implications for studies of correlation (including those published previously), and that previous studies may have reached unreliable conclusions based on a limited set of results, is also new and important.** It appears that we did not express this clearly enough in the original manuscript, and we have now made some changes to the paper to try to resolve this problem.

As far as we can see, the comments of Reviewer #2 were generally positive, and did not include any concerns about the originality of the work. When Reviewer #2 mentions studies "previously presented in the literature", we understand this to be a reference to the conclusion we reach in our paper that the results of such studies should be treated with caution, and not an expression of concern about the originality of our work.

We hope that the comments above, and the changes we have made to the manuscript, clear up any confusion regarding the aims of the paper, its conclusions and their originality. We attach detailed responses to the comments of both reviewers, and a marked-up manuscript showing the changes we have made as a result of those comments.

Yours sincerely,

Jamie Rae, Alex Todd, Ed Blockley and Jeff Ridley

**Responses to anonymous reviewers' comments on *How much should we believe correlations between Arctic cyclones and sea ice extent?* by J.G.L. Rae et al.**

We thank the reviewers for their comments, and give our responses in blue text below.

**Reviewer #1:**

This paper has tried to correlate the sea ice anomalies with the summer cyclonic activity in Arctic and found that the correlations are highly sensitive to the model and resolution used for tracking the cyclonic activity.

While we were indeed looking for correlations between September ice extent and summer Arctic cyclone activity, we went further than this in that we also analysed the robustness or otherwise of such correlations. The sensitivity to model, resolution, and spatial and temporal sampling was the key conclusion of the paper – that the cyclone-ice correlations found by previous authors are not robust because they are sensitive to all these factors.

While the paper is generally well written and the authors explain well their methodology, I do not recommend to accept this paper for publication in TC because this subject has already been discussed in many papers (as well presented by the authors) and that this paper only shows that the variability of sea ice extent is definitely NOT driven by the summer cyclonic activity!

The papers mentioned by the reviewer, which we do indeed cite in our paper, studied such correlations in observations/reanalyses, not in a long coupled GCM run as we did. This meant that they were restricted to the ~30 years of available satellite observations of sea ice extent (as opposed to the 100 years of model output we used). In addition, they were restricted to one resolution and one tracking variable. Those studies found apparent correlations/links between summer cyclones and September ice extent, i.e. they appeared to show that the variability of sea ice extent *was* driven by the summer cyclone activity. We showed that these correlations are dependent on model, resolution, tracking variable, spatial domain, and time period used, and that they are therefore not robust. Our main conclusion was that the results of those previous studies should be treated with caution for this reason.

The correlations found (< 0.44) are not enough relevant to justify a paper in TC and obviously explains why the results are very sensitive to the model and resolution used.

As we explained in the paper, the correlations presented are all significant at least at the 90% level, and often at the 95% level.

If the correlations would be scientifically robust, the sign of the correlations should be for example every time the same which is not the case here. Correlating sea ice extent with summer T850 will give likely better correlations but it is out of the scope of this paper.

This is exactly the point we were trying to make in the paper – that the correlations depend on a range of different factors, and are therefore not robust. Therefore, the correlations which were found in the other studies we cited may also not be robust, and those studies may have drawn misguided conclusions from a limited set of results.

As we mentioned in our previous short response to Reviewer #1, we now believe we did not make the aims of the paper clear enough, leading to a misunderstanding of what those aims were. We have therefore re-worded the abstract and part of the final paragraph of the introduction, and added an extra sentence in the conclusions section. We hope that these changes make the aims of the paper clearer.

**Reviewer #2:**

This paper investigates correlative relationships between Arctic cyclones and September sea ice extent using two different cyclone tracking variables from three different sources (output from two climate model runs and from one reanalysis dataset) . The results show that different tracking variables, model resolution and space/time comparisons can show contrasting cyclone/ice relationships, thereby emphasizing that caution is required when analyzing and interpreting such comparisons (e.g., as previously presented in the literature). This caution is noteworthy and helpful. Thus, this paper is deemed appropriate for TC after considering a few minor suggested revisions as listed below.

P2, L9, Suggest changing 'main ice pack' to 'main pack ice'.
Changed.

P3, L3, Consider starting a new paragraph here.
Done.

P3, L12, Consider rephrasing the following: 'aim of studying correlations between cyclones and Arctic sea ice extent' given Reviewer #1's comment. In other words, specifically state the overall aim of this paper, which is to show how correlations (between cyclones and sea ice extent) depend on sea ice extent, tracking variable, model resolution, and time/space windows used to make the comparison (i.e., what is then stated further down on L15-16).
We have changed this, so that it now reads: "aim of investigating the dependence of cyclone-ice correlations on spatial resolution, tracking variable, and spatial and temporal sampling".

P5, after L4, Consider adding another short paragraph here describing the statistics used to test differences and compute correlations.
We have now added such a paragraph, in new a section (Section 2.4).

P5, L15, Since the first reference to a figure is here, consider moving Fig 5 to Fig 1 and adjust the others accordingly.
Figs 1 to 4 are actually mentioned in the previous sentence ("...track densities and mean intensities (Figs. 1 to 4), as well as the frequency distributions..."), so the figures are already in the correct order.

P5, L30-31, This seems like a general statement summarizing all model/reanalysis comparisons, but then it differs from the last two statements in that paragraph. Reword the first statement to clearly distinguish it from the other points being made (and/or create a table listing these results).
We agree that the description of the results of the Kolmogorov-Smirnov test in the original paper was confusing. We have therefore included a table summarising the results, as suggested by the reviewer, and re-worded the text to try to make it clearer. We have also simplified the discussion by removing the distinction between the 95%, 99% and 99.9% confidence levels, and now simply state whether the frequency distributions can be said to be different with at least 95% confidence.

P6, L14, Consider starting new paragraph here.
Done.

P8, L4, fix typo: 'thesefindings'
Corrected.

Figs 1-4, it would be helpful and of interest to see the difference maps between GC2-N96 and GC2-N216. (This would also be helpful for interpreting the contrasting results between the 2 model runs as presented in Fig 7.)

We have added an extra panel in each of Figs 1-4, and updated the captions accordingly. We have also taken this opportunity to select more-appropriate ranges for the colour scales on the difference plots in those figures. We have added a couple of sentences in Section 3.1 describing what is seen in the new difference maps. Also, because the new maps show GC2-N216 minus GC2-N96, rather than GC2-N96 minus GC2-N216, we have changed the sentence "
[revised manuscript text omitted]